# Effects of *Wnt10a* and *Wnt10b* Double Mutations on Tooth Development

**DOI:** 10.3390/genes14020340

**Published:** 2023-01-28

**Authors:** Kaoru Yoshinaga, Akihiro Yasue, Silvia Naomi Mitsui, Yoshiyuki Minegishi, Seiichi Oyadomari, Issei Imoto, Eiji Tanaka

**Affiliations:** 1Department of Orthodontics and Dentofacial Orthopedics, Tokushima University Graduate School of Biomedical Sciences, 3-18-15 Kuramoto-cho, Tokushima 770-8504, Japan; 2Nakano-Cho niconicoKamKam Dental and Orthodontics, 1-31 Nakano-cho, Tokushima 770-0932, Japan; 3Division of Molecular Medicine, Institute of Advanced Enzyme Research, Tokushima University, 3-18-15 Kuramoto-cho, Tokushima 770-8503, Japan; 4Division of Molecular Biology, Institute of Advanced Enzyme Research, Tokushima University, 3-18-15 Kuramoto-cho, Tokushima 770-8503, Japan; 5Aichi Cancer Center Research Institute, 1-1 Kanokoden Chikusa-ku, Nagoya 464-8681, Japan

**Keywords:** *WNT10A*, *WNT10B*, double mutations, oligodontia, tooth number, functional redundancy

## Abstract

WNT molecules are the regulators of various biological functions, including body axis formation, organ development, and cell proliferation and differentiation. WNTs have been extensively studied as causative genes for an array of diseases. *WNT10A* and *WNT10B*, which are considered to be genes of the same origin, have been identified as causative genes for tooth deficiency in humans. However, the disrupted mutant of each gene does not show a decrease in teeth number. A negative feedback loop, interacting with several ligands based on a reaction–diffusion mechanism, was proposed to be important for the spatial patterning of tooth formation, and WNT ligands have been considered to play a pivotal role in controlling tooth patterning from mutant phenotypes of LDL receptor-related proteins (LRPs) and WNT co-receptors. The *Wnt10a* and *Wnt10b* double-mutants demonstrated severe root or enamel hypoplasia. In *Wnt10a^−/−^* and *Wnt10a^+/−^;Wnt10b^−/−^* mice, changes in the feedback loop may collapse the modulation of fusion or split a sequence of tooth formation. However, in the double-knockout mutant, a decrease in the number of teeth was observed, including the upper incisor or third molar in both jaws. These findings suggest that there may be a functional redundancy between *Wnt10a* and *Wnt10b* and that the interaction between the two genes functions in conjunction with other ligands to control the spatial patterning and development of teeth.

## 1. Introduction

Several genes, such as *EDA*, *MSX1*, *PAX9*, *WNT10A*, and *WNT10B*, have been identified as causative for non-syndromic human tooth agenesis [1,2,3,4,5]. Germline pathogenic variants (GPVs) in each of these causative genes lead to tooth deficiencies, and the sites of tooth defects, produced by each gene, are distinctive. GPVs in *EDA*, *MSX1*, and *PAX9* are majorly responsible for premolar, molar, and anterior tooth deficiencies, respectively. However, the characteristics of defect sites in individuals with *WNT10A* or *WNT10B* GPVs remain unknown [6,7].

WNTs are secretory glycoproteins that play a critical role in the Wnt/β-catenin pathway [8,9]. The Wnt/β-catenin signaling pathway is involved in many aspects of embryonic development and is spatiotemporally activated in the odontogenic regions during all stages of tooth development, suggesting its essential role in tooth formation [10]. *WNT10A* and *WNT10B* are located at chromosome 2q35 and 12q13.12, respectively. *Wnt10a* is preferentially expressed in the dental epithelium and enamel knots during tooth development [11]. *WNT10A* and *WNT10B* are paralogs with 62% amino acid sequence identity in humans and are speculated to have evolved after ancestral gene duplication that occurred in the evolution of the jawed vertebrate lineage [12,13]. The expression patterns of these two genes during tooth development are similar, beginning at the initiation of the dental placode and continuing through the subsequent development stages [14]. According to animal studies, the number of teeth decreases in *Msx1*-, *Pax9*-, and *Eda*-knockout mice, as in the human phenotype. *Msx1^−/−^* mice have palatal clefts and tooth deficiencies [15]. *Pax9^−/−^* mice also have few teeth and a cleft palate [16]. *Eda*^−/−^ mice exhibit tooth agenesis and dwarf teeth [17,18]. However, there are no reports of a decreased number of teeth in *Wnt10a* or *Wnt10b* mutant mice.

*Wnt10* paralogs are expressed at E9.5–E11.5, particularly in limb buds [19]. In the early stages of mouse tooth development, *Wnt10a* and other WNT gene expression, as well as Wnt/β-catenin signaling activity, are primarily restricted to the epithelium [14], and stabilization of Wnt/β-catenin signaling in the embryonic mouse oral epithelium allows for the continued initiation of new tooth development [20]. Dassule and McMahon [14] have elucidated the role of epithelial signaling molecules during early tooth development and found that *Wnt10a* is expressed first in the dental placode and later in primary and secondary enamel knots; therefore, *Wnt10a* potentially functions in tooth initiation and crown patterning. Yamashiro et al. [21] demonstrated that *Wnt10a* is expressed in the secretory odontoblasts, lining dentin, and differentiating odontoblasts around Hertwig’s epithelial root sheath during the later stages of tooth development. Therefore, *Wnt10a* may also potentially function as a key molecule for dentinogenesis during dentin, root, and cusp morphogenesis. *Wnt10b* is also expressed in the tooth germ [22,23]. Expression of *Wnt10b* in mice is first detected at E11.5, a stage showing slight thickening of the presumptive epithelium, which is the major feature of early tooth development. *Wnt10b* is also expressed in the bone marrow, postnatal growth plates [24], and osteoblastic precursors [25].

Heterozygous *WNT10A* GPVs are candidate causes for autosomal-dominant selective tooth agenesis; moreover, homozygous or compound heterozygous are responsible for other autosomal recessive ectodermal dysplasia syndromes such as odonto-onycho-dermal dysplasia (OMIM# 257980) [26]. However, little is known about the mechanism underlying the broad range of human phenotypes in the haploinsufficiency of *WNT10A*. On the other hand, the lateral incisors are reportedly the most frequently missing permanent teeth in patients with *WNT10B* GPVs [22]. Moreover, Kantaputra et al. [7] reported that *WNT10B* GPVs are associated not only with oligodontia and isolated tooth agenesis but also with microdontia, short tooth roots, dental pulp stones, and taurodontism. An in vitro study revealed that mutant *Wnt10b* ligands failed to efficiently induce endothelial differentiation of dental pulp stem cells [22].

Previous studies have not confirmed a reduced number of teeth in *Wnt10a*^−/−^ and *Wnt10b^−/−^* mice. However, *Wnt10a*^−/−^ mice reportedly exhibit abnormal development of ectodermal tissues, such as hair, sweat glands, and taste buds [27]. In place of a reduction in tooth number in *Wnt10a^−/−^* mice, supernumerary teeth were observed behind the mandibular third molars. Moreover, *Wnt10a* deficiency causes enamel hypoplasia, leading to roundish teeth, as well as taurodontic root morphology [28]. In *Wnt10b^−/−^* mice, tooth number and morphogenesis were not evidently delineated, although bone mineral density was reduced [23].

*Wnt10a* and *Wnt10b* are evolutionarily close and show similar expression patterns; therefore, functional redundancy is expected. In this study, various double mutants were generated to verify the functional redundancy of *Wnt10a* and *Wnt10b* in tooth development, resulting in various tooth phenotypes. These findings suggested that *Wnt10a* and *Wnt10b* may compensate for these mutual functions.

## 2. Materials and Methods

### 2.1. Animals

All animal experiments were approved by the Ethics Committee of the Tokushima University for Animal Research (Approval number: T2020-100). All procedures were performed in accordance with the Guidelines for Animal Experiments of the Tokushima University and the ARRIVE guidelines. Both male and female mice were used for morphological tooth observation, and no significant sex differences were noted. Only male mice were used for body length measurement.

### 2.2. Preparation of crRNA–tracrRNA Duplex Ribonucleoprotein (RNP) Complex Assembly

Targeted sequences in *Wnt10a* and *Wnt10b* were designed as shown in Figure 1A,B. crRNAs were purchased from Integrated DNA Technologies (IDT, Coralville, IA, USA, www.idtdna.com/CRISPR-Cas9, accessed on 1 June 2019), in the proprietary Alt-R format. To prepare the RNP complex, two Alt-R crRNAs for each gene were hybridized with Alt-R tracrRNA (catalog number 1072533; IDT) and then assembled with recombinant Cas9 protein (Alt-R S.p. Cas9 nuclease V3, catalog number 1081058; IDT). The resulting RNP complex was diluted to final concentrations of 200, 400, and 50 ng/mL in Opti-MEM I (Life Technologies, Carlsbad, CA, USA) for crRNA, tracrRNA, and Cas9 protein, respectively. After preparation, the samples were used for electroporation.

### 2.3. Mice, Embryo Collection, Electroporation, and Transfer

B6D2F1 (C57BL/6 × DBA2 F1) mice were used to obtain genome-edited F0 mutants in this study. In vitro fertilization (IVF) was performed to obtain fertilized eggs. After a 3 h culture of oocytes and sperm, the eggs were removed from the sperm and cultured for 2 h until electroporation. For preincubation media, mHTF or CARD FERTIUP (KYUDO Co. Ltd., Saga, Japan) was used for each oocyte or sperm. The Genome Editor electroporator and LF501PT1-5 platinum plate electrodes (length: 5 mm, width: 3 mm, height: 0.5 mm, gap: 1 mm) (BEX Co. Ltd., Tokyo, Japan) were used for electroporation. Thirty to forty zygotes, prepared using IVF, were subjected to electroporation at one time. The collected zygotes, cultured in M16 medium, were placed in a line in the electrode gap, filled with 5 μL of Opti-MEM I, containing the RNP complex, and subjected to electroporation. The electroporation conditions were 30 V (3 ms ON +/− 97 ms OFF), for a total of seven times. After electroporation, zygotes were immediately collected from the electrode chamber and cultured in M16 medium at 37 °C and 5% CO_2_ in an incubator. The resulting two-cell embryos were transferred into the oviducts of pseudo-pregnant MCH/ICR mice on the following day.

### 2.4. Stereomicroscope and Micro-Computed Tomography (µCT)

The soft tissue of 4-week-old mice was resected and fixed overnight in 95% ethanol/PBS. The skulls were analyzed using high-resolution µCT (Skyscan 1176, operated at 50 kV and 200 μA; Bruker-microCT, Kontich, Belgium). Two-dimensional images were used to generate three-dimensional (3D) renderings using the CTVox 3D Creator software (version 3.0, Bruker). The resolution of the µCT image was 9 μm/pixel.

### 2.5. Mouse Genotyping

To validate CRISPR/Cas9-mediated mutations, genomic DNA was extracted from the tail biopsies. The genomic regions flanking the guide RNA target were PCR amplified with KOD-Plus-Neo (Toyobo, Osaka, Japan), a *Wnt10a* primer pair (Forward #1 5′ -CAGACTCCCACAAAATGCTTATCCA -3′; Forward #2 5′ -TGCTCTTAGCTCTTAGCC -3′ and reverse 5′ -ATACTTCCTGCCAGCCAGCCCAGGTCT -3′), and a *Wnt10b* primer pair (Forward 5′-CTCCTGTTCTTGGCTTTGTTCAGTC-3’ and reverse 5′-CTGTGTGTGCTGCTGCCCAG-3′), according to the manufacturer’s instructions. After verifying the amplicon size, F0 mice were mated with wild-type C57BL/6J mice, and the disrupted alleles were sequenced with a BigDye Terminator Sequence Kit ver. 3.1 and ABI 3500xL Genetic Analyzer (Applied Biosystems, Foster City, CA, USA) to propagate the alleles of interest. Subsequently, the F2–F4 generations were analyzed.

### 2.6. Statistical Analysis

Each experiment was independently repeated a minimum of three times for each set of conditions. Data are expressed as mean ± standard deviation (SD). Data were analyzed using the Turkey test. * Means *p* < 0.05; ** means *p* < 0.01.

## 3. Results

### 3.1. Generation of Each Wnt10a or Wnt10b Mutant

*Wnt10a* and *Wnt10b* were targeted using the CRISPR/Cas system in mice. The PAM sequence is highlighted in red. The entire *Wnt10a* coding region was deleted in the *Wnt10a*-knockout mutant used in a previous study [28]. In this study, guide RNAs were designed in exon 2 and exon 3 in *Wnt10a*. *Wnt10a* was disrupted from the middle region of exon 2 to the middle region of exon 3, and as a result, *Wnt10a* mutant alleles had 7046-nucleotide deletion (Figure 1A). Endogenous exons 2 to 5 in *Wnt10b* were deleted in the *Wnt10b*^−/−^ mutant used in a previous study [23]. In this study, exons 3 and 4 were targeted in *Wnt10b* mutant alleles, and as a result, 46-nucleotide and 1236-nucleotide deletion occurred (Figure 1B).

### 3.2. Changes in the Number and Shape of Teeth in a Single Mutant of Wnt10a and Wnt10b

Wild-type mice had three molars in the maxilla and mandible: maxillary or mandibular first molar (M1 teeth), second molar (M2 teeth), and third molar (M3 teeth). None of the *Wnt10a*^−/−^ (n = 5) or *Wnt10b*^−/−^ (n = 5) mutant mice generated in this study showed tooth deficiency (Figure 2A,B). Although there were no changes in the number of maxillary teeth in *Wnt10a*^−/−^ mice, 60 percent of *Wnt10a*^−/−^ mutant mice had a supernumerary tooth called fourth molar (M4 teeth) appear behind M3 teeth in the mandible. The M4 tooth was small in size with one cusp (Figure 2B, Table 1). These phenotypes are the same as in previous reports [23,27,28]. In addition, *Wnt10a*^−/−^ mice tooth cusps were flattened, which is consistent with a previous report [27,28] (Figure 2A,B). Their molar had a “rounded-square” crown shape rather than the “rectangular” shape of wild-type molars, and the mesiodistal length of the M1 teeth was decreased (Appendix A). A decrease in the number of cusps compared with that of the wild-type was shown.

The maxillary M1, M2, and M3 teeth in the wild-type mice exhibited three rows of cusps: buccal, lingual, and central. All cusps were posteriorly inclined and separated by deep grooves, buccolingually oriented. Maxillary M1 teeth had three buccal cusps (B1 cusp: Labial anterocone, B2 cusp: Paracone, B3 cusp: Metacone), two lingual cusps (L1 cusp: Anterostyle, L2 cusp: Enterostyle) and three central cusps (cusp 1: Lingual anterocone, cusp 2: Protocone, cusp 3: Hypocone) [29] (Figure 3A). In the mandibular M1, M2, and M3 teeth, the cusps formed by buccal and lingual rows are separated by one or two deep grooves, buccolingually oriented. The mandibular M1 teeth had three buccal cusps (B1 cusp: Labial anteroconid, B2 cusp: Metaconid, B3 cusp: Entoconid), three lingual cusps (L1 cusp: Lingual anteroconid, L2 cusp: Protoconid, L3 cusp: Hypoconid) and one distal cusp (cusp 4: Posteroconid) [29] (Figure 3A). In *Wnt10a*^−/−^ mice, the defect of the L1 cusp in the maxillary M1 and M2 teeth was observed; moreover, the L2 cusp was also defective in the maxillary M2 and M3 teeth (Figure 3B,C). In the mandible, the L1 cusp in M1 teeth, cusp 4 in M2 teeth, and the L3 cusp in M3 teeth were not formed (Figure 3B,C). In contrast, there were no changes in the number and morphology of teeth in *Wnt10b*^−/−^ mice (Figure 2A,B). No changes in the cusp pattern in *Wnt10b*^−/−^ mice were observed (Figure 3B,C).

### 3.3. Variations in the Number and Shape of Teeth in Wnt10a and Wnt10b Double Mutants

The double heterozygous mutant of *Wnt10a* and *Wnt10b* did not show any differences in craniofacial development or the number and shape of teeth compared with the wild-type mice (Figure 2C,D). No differences in the molar cusp pattern in *Wnt10a*^+/−^;*Wnt10b*^+/−^ mice were observed (Figure 3B,C). In *Wnt10a*^−/−^;*Wnt10b*^+/−^ mice, the tooth shape was flattened in both the maxilla and the mandible, and 50% of *Wnt10a*^−/−^;*Wnt10b*^+/−^ mice had the ectopic molar M4 teeth behind M3 teeth in the mandible as shown in *Wnt10a*^−/−^ mice (Figure 2C,D, Table 1). The rounded molar cusp pattern was also the same as *Wnt10a*^−/−^ mice; the L1 cusps were not formed in the maxillary M1 teeth, and furthermore, M2 teeth lost the L1 and L2 cusps. In M3 teeth, the L2 cusps were defect (Figure 3B,C). In *Wnt10a*^+/−^;*Wnt10b*^−/−^ mice, the tooth shape was slightly round, and a supernumerary tooth, with a single cusp and root, appeared anterior to the mandibular M1 teeth in the majority of cases, and 80% of *Wnt10a*^+/−^;*Wnt10b*^−/−^ mice had the supernumerary tooth (Table 1). Although there were variations in the size of the supernumerary tooth, it was smaller than the mandibular M1 teeth (arrowhead, Figure 2C,D). In contrast, the maxillary molars of *Wnt10a*^+/−^;*Wnt10b*^−/−^ mice differed in size, that is to say, the M2 teeth were larger than the M1 teeth (Appendix A). The maxillary M1 teeth of *Wnt10a*^+/−^;*Wnt10b*^−/−^ mice were smaller than that in the wild-type mice (Figure 2C,D, Appendix A). In the maxillary M1 teeth, the B2 cusp and the B3 cusp were defect; moreover, the L1 cusp and the L2 cusp were fused in one cusp. An excessive cusp appeared nearby the B2 cusp in the maxillary M2 teeth. In the mandible, the L1 cusp was defective in the M1 teeth. In *Wnt10a*^−/−^;*Wnt10b*^−/−^ mice, the molar tooth size was significantly small, and the M3 teeth were missing or dwarf M3 teeth in the maxilla and mandible (Figure 2C,D, Appendix A). The L1 and L2 cusps were defect in the maxillary M1 and M2 teeth. In the mandible, the L1 cusp in M1 teeth and the L2 and L3 cusps in M2 teeth were defective. If there were dwarf M3 teeth, three cups were fused in one cusp (Figure 3B,C). The maxillary incisors were congenitally missing, and the mandibular incisors were thinner and grew longer owing to missing maxillary incisors (Figure 2C,D).

### 3.4. Only Double Homozygous Alleles of Wnt10a and Wnt10b Show Developmental Retardation

The body length of *Wnt10a* and *Wnt10b* double mutants, except for *Wnt10a*^−/−^;*Wnt10b*^−/−^ mice, showed no difference compared with the wild-type mice and appeared normal (Figure 4A). At 4 weeks, *Wnt10a*^−/−^ and *Wnt10b*^−/−^ mice showed a lower growth rate than the wild-type mice. Only *Wnt10a*^−/−^;*Wnt10b*^−/−^ mice showed significant (*p* < 0.05) differences in body length (Figure 4A,B).

### 3.5. Root Morphology of Wnt10a and Wnt10b Mutant Mice

For wild-type mice, each maxillary molar had three roots: the mesial buccal root (MB root), distal buccal root (DB root), and palatal root (P root) (Figure 5A,B). In *Wnt10a*^−/−^ mice, the mesial buccal and palatal roots were fused in the maxillary M1 and M2 teeth, and the maxillary M3 teeth showed a single root (arrow, Figure 5A). No differences in the number and pattern of maxillary molar roots were observed in *Wnt10b*^−/−^, *Wnt10a*^+/−^;*Wnt10b*^+/−^, and *Wnt10a*^+/−^;*Wnt10b*^−/−^ mice (Figure 5A). In *Wnt10a*^−/−^;*Wnt10b*^+/−^ mice, the maxillary M1 and M2 teeth showed the same root patterns as observed in *Wnt10a*^−/−^ mice. In contrast, all maxillary molars had a single root, and root bifurcation was not observed in *Wnt10a*^−/−^;*Wnt10b*^−/−^ mice (Figure 5A).

For the mandible, the M1 and M2 teeth of wild-type mice had two roots, the mesial (M root) and distal root (D root), and the M3 teeth had a single root. As reported in previous studies [28], in *Wnt10a*^−/−^ mouse molars, enlargement of the pulp chamber was observed, resulting in taurodonts. *Wnt10a*^−/−^;*Wnt10b*^+/−^ mice showed more severe taurodontism phenotypes with an apical displacement of the pulp floor and bifurcation of the roots compared to *Wnt10a*^−/−^ mice. No differences in root morphogenesis were observed between the *Wnt10b*^−/−^ and *Wnt10a*^+/−^;*Wnt10a*^+/−^ mice and wild-type mice. In *Wnt10a*^+/−^;*Wnt10b*^−/−^ mice, the root morphology of the M1, M2, and M3 teeth were not changed compared to wild-type mice; however, the supernumerary tooth with a single root in diastema appeared (asterisk, Figure 5B). In *Wnt10a*^−/−^;*Wnt10b*^−/−^ mice, the mesial and distal roots were fused in one root, showing taurodonts (Figure 5B).

## 4. Discussion

### 4.1. WNT Signaling Regulates Tooth Development

WNT signaling regulates the patterning and differentiation of many tissues and organs in the developing embryo through the temporary and spatial restricted expression of multi-gene families encoding 18 ligands, receptors, pathway modulators, and intracellular components. WNTs include their paralogs in family members [13]. They are the result of gene duplication. Therefore, similar expression patterns are shown [19]. The deregulation of Wnt/β-catenin signaling causes many diseases [5,6,7,11]. The WNT ligands cooperate with many factors. Wise, also called USAG-1, Sostdc1, or Ectodin, is the dual antagonist to Wnt and BMP (bone morphogenetic protein) signaling, and Lrp4 is a negative Wnt co-receptor. Wnt/β-catenin signaling affects the development of teeth and other ectodermal appendages and controls the number and formation of the tooth including cusps and roots [9].

In mice, there are one incisor and three molars in each jaw quadrant. A gap between the incisor and the molars is called diastema. The diastema normally lacks the function of tooth formation because Wise suppresses the survival of diastemal buds [8,30]. Both diastemal tooth buds in the maxilla, called MS and R2, disappear by apoptosis. The anterior bud regresses in the mandible; however, the first molar in the mandible absorbs the posterior bud, also called R2 [31]. These tooth buds were considered as the rest of the premolars. Remarkably, supernumerary teeth such as premolars were reported in *Wise*^−/−^ or *Lrp4* mutants [30,32], which suggest the activation of Wnt/β-catenin signaling leads to the survival of R2 and controls the balance of tooth field repartition. Tooth number or the developmental repartition in *Lrp4* mutants, with *Lrp5* or *Lrp6* mutants, have recovered normal dentition, suggesting that Lrp5/6 is a Wnt co-receptor competing with Lrp4 for binding to Frizzled receptors. Several scenarios regarding the relationship between Lrp4, Lrp5/6, Wise, and Wnt have been described, and it has been hypothesized that Lrp4 and Wise physically interact to negatively regulate Wnt/β-catenin signaling [8,31] Ectopic expression of *Lrp4* leads to the disruption of the normal distribution and/or function of *Wise*, causing an elevated Wnt/β-catenin signaling at the epithelial signaling center of R2 [32]. In this study, *Wnt10b*^−/−^ mice did not show any phenotypes of tooth number, root, and molar cusp pattern; however, *Wnt10a*^+/−^;*Wnt10b*^−/−^mice had the supernumerary tooth in diastema, indicating that changes in quantity of Wnt signaling cause survival of R2.

### 4.2. Tooth Morphology and Number Are Affected in Several Double Mutants

The double mutants of *Wnt10a* and *Wnt10b* generated in this study showed various tooth phenotypes. Although, the double heterozygous mutant *Wnt10a*^+/−^;*Wnt10b*^+/−^ mice did not exhibit any differences in maxillofacial morphology, number of teeth, and tooth morphology. *Wnt10a*^−/−^;*Wnt10b*^+/−^ mice showed flattened cusps, defective root bifurcation, and a fourth molar (M4 teeth) in the mandible, as shown in *Wnt10a* single mutant mice; however, the flatness of cusps and taurodontism was considerably more severe than those in *Wnt10a*^−/−^ mice. As both *Wnt10a* and *Wnt10b* are expressed in enamel knots and odontoblasts, these two genes may compensate for their roles in odontogenesis. In contrast, tooth number did not change in *Wnt10b*^−/−^mice. However, *Wnt10a*^+/−^;*Wnt10b*^−/−^ mice exhibited a supernumerary tooth in mandibular diastema and size differentiation of the maxillary M1 and M2 teeth.

*Wise*^−/−^ or *Lrp4* mutants had supernumerary teeth located in the diastema region mesial to M1 teeth [30,32]. In this study, *Wnt10a*^+/−^;*Wnt10b*^−/−^ mice had supernumerary teeth ahead of M1 teeth in diastema, suggesting that a similar molecular mechanism to *Wise*^−/−^ or *Lrp4* mutants may have occurred. However, the extra teeth of *Wnt10a*^−/−^ and *Wnt10a*^−/−^;*Wnt10b*^+/−^ mice are located behind the M3 teeth, which suggests that the underlying mechanism of supernumerary tooth formation in *Wnt10a*^−/−^ and *Wnt10a*^−/−^;*Wnt10b*^+/−^ mice might be different from that of other mutant mouse models. It is possible that the functional redundancy of *Wnt10a* and *Wnt10b* may control spatial patterning and the development of teeth.

### 4.3. Growth Rate Is Decreased in Wnt10a and Wnt10b Double Mutants

Both *Wnt10a* and *Wnt10b* stimulate osteoblastogenesis via a b-catenin-dependent mechanism [33]. *Wnt10a*-knockout mice show significant bone loss [34]. Alveolar bone density was also decreased in *Wnt10a*^−/−^ mice. Therefore, *Wnt10a* is considered to be important for the maintenance of postnatal bone volume [27]. *Wnt10b* expression is observed in the bone marrow. *Wnt10b*-single-knockout mice also show loss of bone volume [23]. However, single mutants of *Wnt10a* or *Wnt10b* appear to have normal growth rates. Although *Wnt10a*^+/−^;*Wnt10b*^+/−^, *Wnt10a*^−/−^;*Wnt10b*^+/−^ and *Wnt10a*^+/−^;*Wnt10b*^−/−^ mice showed normal growth rates, only the double-knockout mutant *Wnt10a*^−/−^;*Wnt10b*^−/−^ mice showed a marked decrease in body length and a weak appearance. This finding suggests that the cooperation of both *Wnt10a* and *Wnt10b* maintains systemic bone homeostasis.

### 4.4. Wnt10a and Wnt10b Function Have a Great Effect on Repartition of Tooth Number

Mice with knockouts of *MSX1*, *PAX9*, and *EDA*, which are the causative genes of oligodontia in humans, demonstrate developmental arrest of tooth formation or a decrease in tooth number [15,16]. Patients with GPVs in *WNT10A* and *WNT10B* show tooth deficiencies; however, none of the knockout mutants have displayed a decrease in tooth number. Regarding other WNT family genes, *Wnt5a* is expressed in the teeth and palate. Although *WNT5A* GPVs have not been reported in human disease, *Wnt5a*^−/−^ mice die a few hours after birth because of the cleft palate phenotype. The incisors and molars observed in these cases are reduced in size, with abnormally patterned cusps at P0 [35]. *Wnt10a*^−/−^, *Wnt10a*^−/−^;*Wnt10b*^+/−^ and *Wnt10a*^+/−^;*Wnt10b*^−/−^ mice exhibited increased tooth numbers, such as M4 teeth behind the M3 teeth in the mandible or supernumerary teeth in diastema. The spatial patterning of teeth, characterized by the size and number of teeth and their cusps, has been described using a reaction–diffusion mechanism [36].

Cho et al. [37] proposed a new reaction–diffusion model, the Wnt-Shh-Sostdc1 negative feedback loop, comprising an activator, mediator, and inhibitor as Wnt, Shh, and Sostdc1, respectively, for the spatial patterning of teeth. Wnt signaling induces Shh, which leads the Wnt/b-catenin pathway to be suppressed by Shh indirectly via Sostdc1. It is reported that Bmp induction of Sostdc1 expression was disturbed by Shh [38]. Moreover, the reduction of Shh signaling causes severe fused molar in *Sostdc1*^−/−^ mice, which suggests that Shh affects preventing molar fusion downstream of *Sostdc1*. In addition, it was indicated that Shh works as a Wnt signaling antagonist via the upregulation of *Dkk1* [8].

*Wnt10a*^−/−^;*Wnt10b*^−/−^ mice showed reduced size or the number of teeth. These phenotypes are very similar to Wnt-inhibitor *Dkk1* overexpressed mice. The mandibular molars of Dkk1 overexpressed mice had significantly flattened and small cusps [9]. This might suggest that the phenotypes of *Wnt10a*^−/−^;*Wnt10b*^−/−^ mice are affected by the Wnt-Shh-Sostdc1 negative feedback loop. Tabby, the *Eda* mutant mice also shows similar tooth phenotypes to *Wnt10a*^−/−^;*Wnt10b*^−/−^ mice including smaller tooth and fewer cusps forming [18]. *Eda* is a target gene of Wnt/β-catenin signaling, and Eda signaling may modulate multiple components of the integrated signaling network that regulates the balance of activators and inhibitors of tooth morphogenesis and patterning. Therefore, it is possible that severe signaling reduction may occur in *Wnt10a* and *Wnt10b* double mutants.

Our study highlights the importance of both *Wnt10a* and *Wnt10b* functions in the determination or repartition of tooth number. The findings from this study may prove to be of significance in understanding the feedback loop mechanism involved in the spatial patterning of teeth.

## Figures and Tables

**Figure 1 genes-14-00340-f001:**
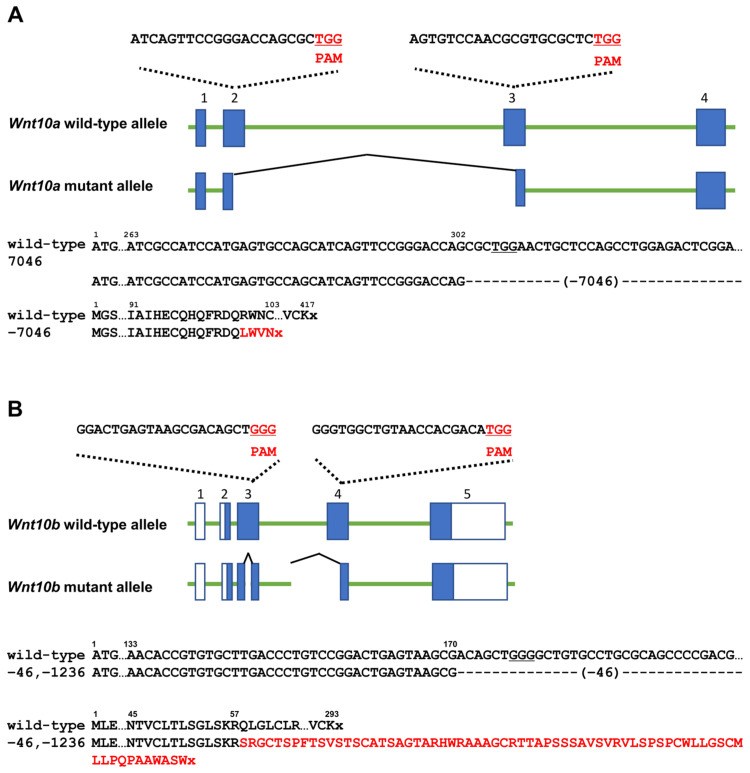
*Wnt10a* and *Wnt10b* targeting using CRISPR/Cas system. Targeted regions in mouse *Wnt10a* (**A**) and *Wnt10b* (**B**) for genome editing. The PAM sequence is highlighted in red. The predicted consequences of the mutation on amino acid sequences are highlighted in red. The stop codons are shown as x.

**Figure 2 genes-14-00340-f002:**
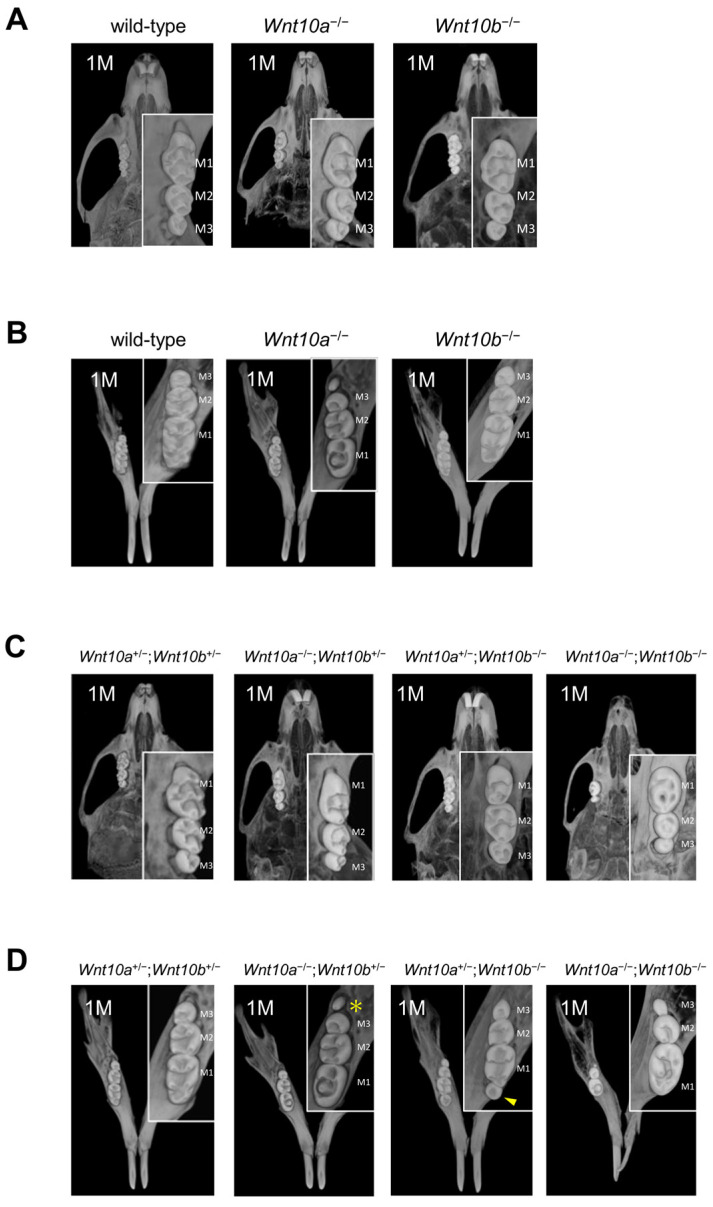
Representative tooth phenotypes in 4-week-old wild-type and mutant mice. µCT analysis of maxillae (**A**,**C**) and mandibles (**B**,**D**) from wild-type (control), *Wnt10a*^−/−^, *Wnt10b*^−/−^, *Wnt10a*^+/−^;*Wnt10b*^+/−^, *Wnt10a*^−/−^;*Wnt10b*^+/−^, *Wnt10a*^+/−^;*Wnt10b*^−/−^, and *Wnt10a*^−/−^;*Wnt10b*^−/−^ mice at 4 weeks. M1, first molar; M2, second molar; M3, third molar. The arrowhead and asterisk indicate a supernumerary tooth behind mandibular M3 and anterior to mandibular M1, respectively.

**Figure 3 genes-14-00340-f003:**
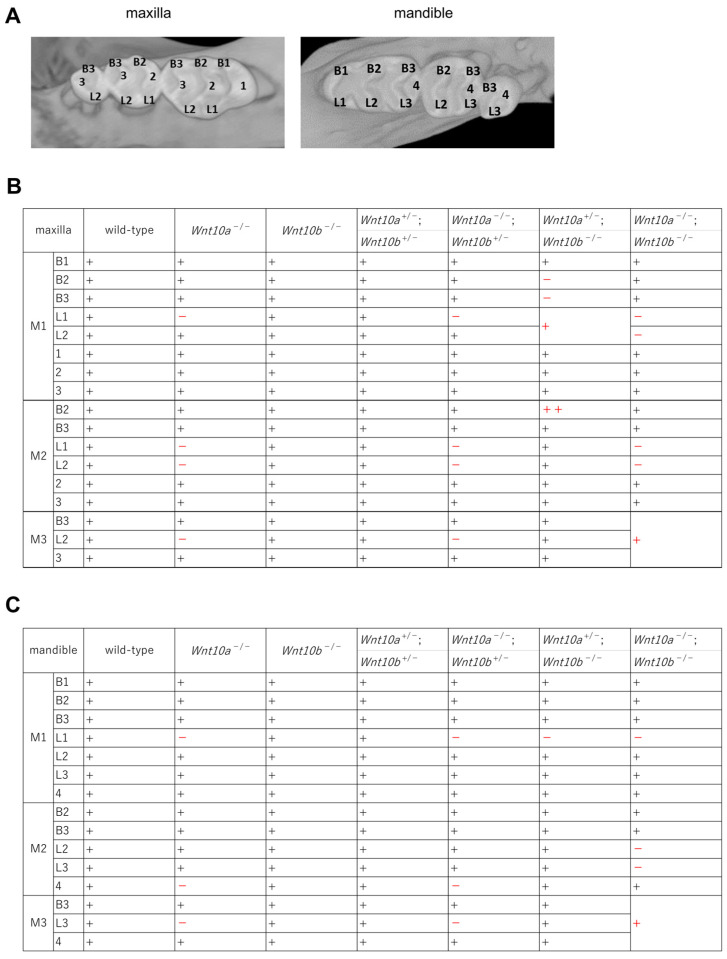
Crown morphology of 4-week-old wild-type and mutant mice. (**A**) Buccal cusps (B1, B2, and B3), lingual cusps (L1, L2, and L3), and central cusps (1, 2, 3, and 4) are counted from mesial to distal molar in wild-type mice. Summarizing mutant mice phenotypes in the maxilla (**B**) and mandible (**C**). +, − and ++ indicate complete, missing, and extra cusps, respectively. Morphologically changed cusps are highlighted in red.

**Figure 4 genes-14-00340-f004:**
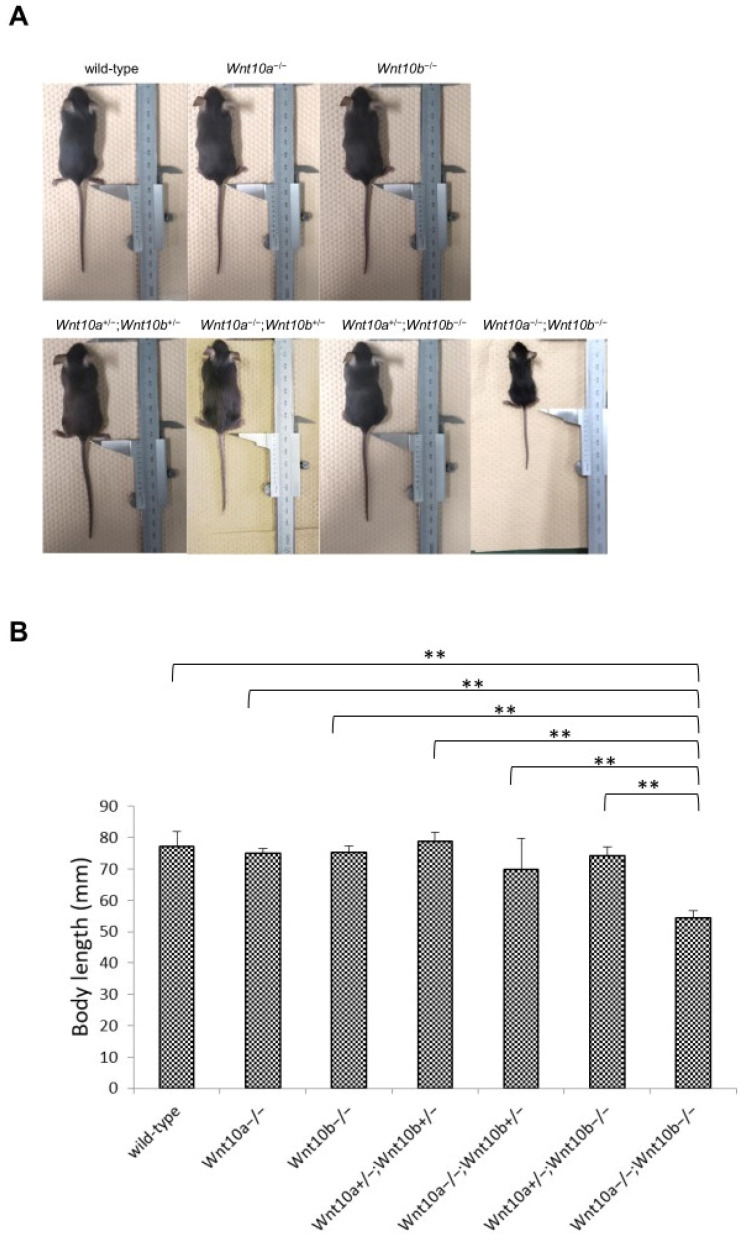
*Wnt10a* and *Wnt10b* double mutant mice have different growth rates. (**A**) Gross phenotype of 4-week-old wild-type and mutant mice. (**B**) In mutant mice, only *Wnt10a*^−/−^;*Wnt10b*^−/−^ mice showed a lesser growth rate than that of wild-type mice at 4 weeks. ** means *p* < 0.01.

**Figure 5 genes-14-00340-f005:**
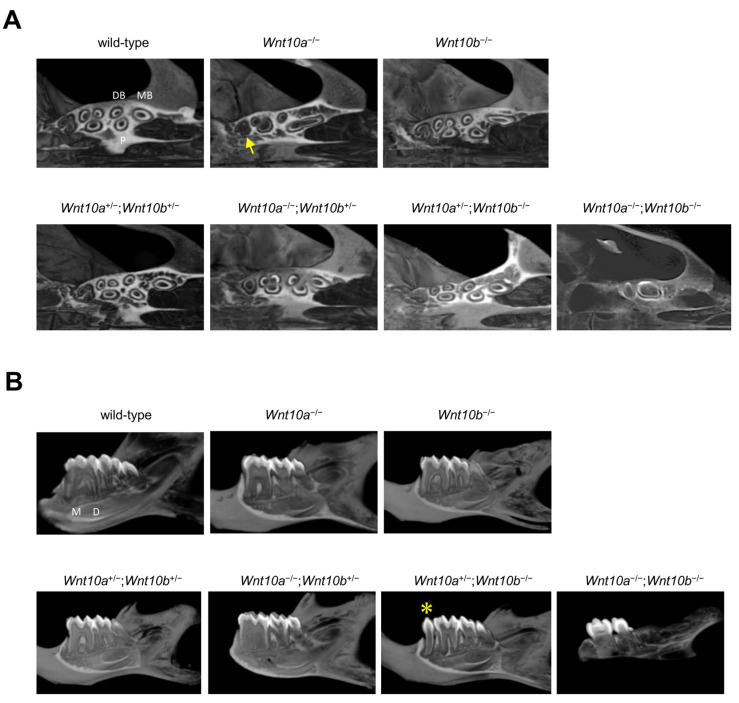
Molar root morphology of 4-week-old wild-type and mutant mice. Horizontal (**A**) and sagittal sections (**B**) of the right maxillary molars of wild-type and mutant mice. The arrow indicates a single root in maxillary M3, and the asterisk indicates a supernumerary tooth in diastema.

**Table 1 genes-14-00340-t001:** Tooth phenotypes of *Wnt10a* and *Wnt10b* mutant mice. Ratio means the frequencies of distinct phenotypes in each genotype.

Genotypes	Phenotypes	Ratio
*Wnt10a* ^−/−^	M4 appearance	3/5
*Wnt10b* ^−/−^	No change	5/5
*Wnt10a*^+/−^;*Wnt10b*^+/−^	No change	10/10
*Wnt10a*^−/−^;*Wnt10b*^+/−^	M4 appearance	4/8
*Wnt10a*^+/−^;*Wnt10b*^−/−^	A supernumerary tooth appearance	8/10
*Wnt10a*^−/−^;*Wnt10b*^−/−^	M3 defect	3/5

## Data Availability

Not applicable.

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
