# Peer review of "Effects of Wnt10a and Wnt10b Double Mutations on Tooth Development"

_genes, 2023, doi:10.3390/genes14020340_

Round 1
Reviewer 1 Report
It is an interesting manuscript, it presents an innovative methodology regarding gene editing (CRISPR-Cas9), as well as good background support in the introduction regarding the function of wnt family members.
Observations
The t test is not an adequate statistic test considering that it has 7 groups including a control (wild-type). An analysis of normality should be done to determine which multiple comparison test could work better (Tukey or Dunn maybe), in addition to not mentioning statistical significance. The asterisk in Fig 4b or in the graphs in appendix 1 must be explained if it is representing significance, including what p value was considered.
The results on dental, cusp and root morphology are very well documented, however, alveolar bone density from µCT analysis is poor. The idea of the “Paralog” gene is somewhat lost, if the Wnta and Wntb genes were generated by duplication their proteins, these could be similar but not identical functions; and this part should be further exploited in bone homeostasis analysis considering the background for Wntb.
The main problem is the redaction of the text, ideas are abundant regarding the morphological data, however, when establishing a relationship with the genotype, certain occasions lead to contradictions and redundancies, particularly in the discussion (section 4.2).
Section 4.3 of the discussion was descriptive of the results, and non-confrontation was established to support that the wnta-/- and wntb-/- condition affect bone homeostasis.
The final conclusion about "" is difficult to understand; if you intent to mention that your results impact over affection of feedback loop (understanding wnt-shh-sostdc1), is partial, because this manuscript have no evidence about shh or sosdc1, or maybe the redaction disturb the idea.
Author Response
We thank you for careful reading our manuscript and for giving useful comments. I am returning herewith a manuscript revised according to your comments.
Observations
The t test is not an adequate statistic test considering that it has 7 groups including a control (wild-type). An analysis of normality should be done to determine which multiple comparison test could work better (Tukey or Dunn maybe), in addition to not mentioning statistical significance. The asterisk in Fig 4b or in the graphs in appendix 1 must be explained if it is representing significance, including what p value was considered.
⇒Analysis is performed again by Tukey test. We changed Fig 4b and appendix 1. Moreover, we added statistical significance to figure legend of Fig 4b and appendix 1
The sentence in section 2.4 “Data were analyzed using the two-tailed Student's t-test” is replaced with “Data were analyzed using the Tukey test. * means p<0.05. ** means p<0.01 ”.
The results on dental, cusp and root morphology are very well documented, however, alveolar bone density from µCT analysis is poor. The idea of the “Paralog” gene is somewhat lost, if the Wnta and Wntb genes were generated by duplication their proteins, these could be similar but not identical functions; and this part should be further exploited in bone homeostasis analysis considering the background for Wntb.
⇒Wnt10a showed not only alveolar bone loss but also trabecular bone defect in ref 34. The alveolar bone loss was investigated at 12-months-old Wnt10a-/- mice. For Wnt10b-/- mice, the bone lost femur was shown in ref 23, on the other hand, it is unclear if Wnt10b-/- mice show alveolar bone loss at old age. The authors of ref 23 simply might not pay attention to the alveolar bone. On the other hand, Wnt10a-/-;Wnt10b-/- mice was small in size already at birth (data not shown). So Wnt10a and Wnt10b cooperate for systemic bone development from embryonic stages.
The main problem is the redaction of the text, ideas are abundant regarding the morphological data, however, when establishing a relationship with the genotype, certain occasions lead to contradictions and redundancies, particularly in the discussion (section 4.2).
⇒We modified this part and added the following text in section 4.2 .
Wise-/- or Lrp4 mutants had supernumerary teeth located in diastema region mesial to M1 teeth [30,32]. These mutant mice explain that a complexed of various signaling decides tooth morphology and number [33]. In this study, Wnt10a+/-;Wnt10b-/- mice had supernumerary teeth ahead of M1 teeth in diastema, suggesting that similar molecular mechanism to Wise-/- or Lrp4 mutants may be occurred. However, the extra teeth of Wnt10a-/- and Wnt10a-/-;Wnt10b+/- mice are located behind M3 teeth, which suggests that the underlying mechanism of supernumerary tooth formation in Wnt10a-/- and Wnt10a-/-;Wnt10b+/- mice might be different from that of other mutant mouse models. It is possible that the functional redundancy of Wnt10a and Wnt10b may control spatial patterning and development of teeth.
Section 4.3 of the discussion was descriptive of the results, and non-confrontation was established to support that the wnta-/- and wntb-/- condition affect bone homeostasis.
⇒The sentence “This finding suggests that Wnt10a and Wnt10b maintain bone homeostasis” in section 4.3 is replaced with “This finding suggests that the cooperation of both Wnt10a and Wnt10b maintain systemic bone homeostasis”.
The final conclusion about "" is difficult to understand; if you intent to mention that your results impact over affection of feedback loop (understanding wnt-shh-sostdc1), is partial, because this manuscript have no evidence about shh or sosdc1, or maybe the redaction disturb the idea.
⇒We have added the following text to Discussion part and modified this part (section 4.4).
Wnt10a-/-;Wnt10b-/- mice showed reduced size or number of teeth. These phenotypes are very similar to Wnt-inhibitor Dkk1 overexpressed mice. Mandibular molars of Dkk1 overexpressed mice had significantly flattened and small cusps. These might suggest that the phenotypes of Wnt10a-/-;Wnt10b-/- are concerned by Wnt-Shh-Sostdc1 negative feedback loop.

Reviewer 2 Report
Comments and Suggestions for Authors
The paper the effect of WNT10A and WNT10B Knockout on the number and morphology of teeth and suggests the presence of functional redundancy between the two genes.
Introduction:
Page 2, line 75:
Heterozygous WNT10A GPVs are candidate causes for autosomal-dominant selective tooth agenesis, however, homozygous or compound heterozygous are responsible for other autosomal recessive ectodermal dysplasia syndromes.
Materials and Methods:
· Page 3, Lines (109-114):
The paragraph “We used wild-type mice, single mutant mice (Wnt10a-/-, Wnt10b-/-) and double 109 mutant mice (Wnt10a+/-;Wnt10b+/-, Wnt10a-/-;Wnt10b+/-, Wnt10a+/-;Wnt10b-/-, Wnt10a-110 /-;Wnt10b-/-). Frequencies of distinct phenotypes in each genotype were also shown in 111 Table 1.” and table 1 are better transferred from the Materials and Methods section to the results section.
· Page 3, Lines 109:
The legend of table 1 should describe what the ratio in the right column of the table denotes.
Results:
· Page 5, line (188-189):
“Their molar had a rounded-square crown shape than the rectangular shape of wild-type molars”, please add “rather” before “than”.
· Page 6, line 207:
“and distal cusp” should be replaced with “and one distal cusp”.
· Page 6, line 209:
“Moreover L2 cusp was also defect in maxillary M2 and M3 teeth”, please replace “defect” with “defective”.
· Page 7, lines (212-213):
The sentence “Any changes of the cusp pattern 212 in Wnt10b-/- mice was not observed” should be replaced with “No changes of the cusp pattern in Wnt10b-/- mice were observed”.
· Page 7, lines (223-224):
The sentence “Any differences of the molar cusp pattern in Wnt10a+/-;Wnt10b+/- mice were not observed” should be replaced with “No difference of the molar cusp pattern in Wnt10a+/-;Wnt10b+/- 223 mice was observed”.
· Page 8, line 229:
The phrase “wad defect” should be replaced with “was defective”
· Page 8, line 239:
The phrase “L1 cusp was defect M1 teeth” should be replace with “L1 cusp was defective in M1 teeth”.
· Page 8, line 240:
The phrase “M3 teeth was missing” should be replaced with “M3 teeth were missing”.
· Page 8, line 243:
The word “defect” should be replaced with “defective”.
Discussion:
· Page 10, lines (284-193):
The whole paragraph does not have any references.
· Page 11, lines (301-302):
The sentence “The diastema is normally lose the function of tooth formation” should be replaced with “The diastema normally lacks the function of tooth formation”.
· Page 11, lines (309-310):
The sentence “a reduction in dentin formation and missing of dwarf teeth was observed” should be replaced with “a reduction in dentition formation and missing or dwarf teeth were observed”.
· Page 11, line 339:
“-catenin-dependent” is missing “β”.
Author Response
We thank you for careful reading our manuscript and for giving useful comments. I am returning herewith a manuscript revised according to your comments.
Introduction:
Page 2, line 75:
Heterozygous WNT10A GPVs are candidate causes for autosomal-dominant selective tooth agenesis, however, homozygous or compound heterozygous are responsible for other autosomal recessive ectodermal dysplasia syndromes.
⇒The sentence “Multiple studies have reported that heterozygous WNT10A GPVs are candidate causes for autosomal-dominant selective tooth agenesis, including the maxillary lateral incisor, and other autosomal recessive ectodermal dysplasia syndromes, such as odonto-onycho-dermal dysplasia (OMIM# 257980) [26].” is replaced with “Heterozygous WNT10A GPVs are candidate causes for autosomal-dominant selective tooth agenesis, moreover, homozygous or compound heterozygous are responsible for other autosomal recessive ectodermal dysplasia syndromes such as odonto-onycho-dermal dysplasia (OMIM# 257980) [26]”.
Materials and Methods:
- Page 3, Lines (109-114):
The paragraph “We used wild-type mice, single mutant mice (Wnt10a-/-, Wnt10b-/-) and double 109 mutant mice (Wnt10a+/-;Wnt10b+/-, Wnt10a-/-;Wnt10b+/-, Wnt10a+/-;Wnt10b-/-, Wnt10a-110 /-;Wnt10b-/-). Frequencies of distinct phenotypes in each genotype were also shown in 111 Table 1.” and table 1 are better transferred from the Materials and Methods section to the results section.
⇒The paragraph “We used wild-type mice, single mutant mice (Wnt10a-/-, Wnt10b-/-) and double 109 mutant mice (Wnt10a+/-;Wnt10b+/-, Wnt10a-/-;Wnt10b+/-, Wnt10a+/-;Wnt10b-/-, Wnt10a-110 /-;Wnt10b-/-). Frequencies of distinct phenotypes in each genotype were also shown in 111 Table 1” are deleted. Table 1 are transferred from the Materials and Methods section to the results section.
- Page 3, Lines 109:
The legend of table 1 should describe what the ratio in the right column of the table denotes.
⇒The sentence “Ratio means frequencies of distinct phenotypes in each genotype” is added in the legend of table 1.
Results:
- Page 5, line (188-189):
“Their molar had a rounded-square crown shape than the rectangular shape of wild-type molars”, please add “rather” before “than”.
⇒The word “rather” is added before “than”.
- Page 6, line 207:
“and distal cusp” should be replaced with “and one distal cusp”.
⇒The word “and distal cusp” is replaced with “and one distal cusp”.
- Page 6, line 209:
“Moreover L2 cusp was also defect in maxillary M2 and M3 teeth”, please replace “defect” with “defective”.
⇒The word “defect” is replaced with “defective”.
- Page 7, lines (212-213):
The sentence “Any changes of the cusp pattern 212 in Wnt10b-/- mice was not observed” should be replaced with “No changes of the cusp pattern in Wnt10b-/- mice were observed”.
⇒The sentence “Any changes of the cusp pattern 212 in Wnt10b-/- mice was not observed” is replaced with “No changes of the cusp pattern in Wnt10b-/- mice were observed”.
- Page 7, lines (223-224):
The sentence “Any differences of the molar cusp pattern in Wnt10a+/-;Wnt10b+/- mice were not observed” should be replaced with “No difference of the molar cusp pattern in Wnt10a+/-;Wnt10b+/- 223 mice was observed”.
⇒The sentence “Any differences of the molar cusp pattern in Wnt10a+/-;Wnt10b+/- mice were not observed” is replaced with “No difference of the molar cusp pattern in Wnt10a+/-;Wnt10b+/- mice was observed”.
- Page 8, line 229:
The phrase “wad defect” should be replaced with “was defective”
⇒The phrase “wad defect” is replaced with “was defective”.
- Page 8, line 239:
The phrase “L1 cusp was defect M1 teeth” should be replace with “L1 cusp was defective in M1 teeth”.
⇒The phrase “L1 cusp was defect M1 teeth” is replaced with “L1 cusp was defective in M1 teeth”.
- Page 8, line 240:
The phrase “M3 teeth was missing” should be replaced with “M3 teeth were missing”.
⇒The phrase “M3 teeth was missing” is replaced with “M3 teeth were missing”.
- Page 8, line 243:
The word “defect” should be replaced with “defective”.
⇒The word “defect” is replaced with “defective”.
Discussion:
- Page 10, lines (284-193):
The whole paragraph does not have any references.
⇒Appropriate references are added in this paragraph.
- Page 11, lines (301-302):
The sentence “The diastema is normally lose the function of tooth formation” should be replaced with “The diastema normally lacks the function of tooth formation”.
⇒The sentence “The diastema is normally lose the function of tooth formation” is replaced with “The diastema normally lacks the function of tooth formation”.
- Page 11, lines (309-310):
The sentence “a reduction in dentin formation and missing of dwarf teeth was observed” should be replaced with “a reduction in dentition formation and missing or dwarf teeth were observed”.
⇒The sentence “a reduction in dentin formation and missing of dwarf teeth was observed” is replaced with “a reduction in dentition formation and missing or dwarf teeth were observed”.
- Page 11, line 339:
“-catenin-dependent” is missing “β”.
⇒The word “β” is added in front of “-catenin-dependent”.

Round 2
Reviewer 1 Report
The manuscript showed better presentation and more correlation in the discussion redaction of their results.
Author Response
We thank you for careful reading our manuscript again. I am returning herewith an improved manuscript of several typos. I hope that the revised paper meets your approval and will be more suitable for publication in Genes.
319: morphology, Wnt10a-/-;Wnt10b+/- mice
⇒ morphology. Wnt10a-/-;Wnt10b+/- mice
371: These might suggest that the phenotypes of Wnt10a-/-;Wnt10b-/- are concerned by Wnt-Shh-Sostdc1 negative feedback loop.
⇒ This might suggest that the phenotypes of Wnt10a-/-;Wnt10b-/- mice are concerned by Wnt-Shh-Sostdc1 negative feedback loop.
The size of the figures was totally modified for easily viewing.